# A Review on the Applications of Natural Biodegradable Nano Polymers in Cardiac Tissue Engineering

**DOI:** 10.3390/nano13081374

**Published:** 2023-04-15

**Authors:** Rabia Aziz, Mariarosaria Falanga, Jelena Purenovic, Simona Mancini, Patrizia Lamberti, Michele Guida

**Affiliations:** 1Department of Information and Electrical Engineering and Applied Mathematics (DIEM), University of Salerno, 84084 Fisciano, Italy; mfalanga@unisa.it (M.F.); smancini@unisa.it (S.M.); plamberti@unisa.it (P.L.); miguida@unisa.it (M.G.); 2Consiglio Nazionale Delle Ricerche (CNR)-Istituto Officina dei Materiali (IOM), Area Science Park Basovizza S.S. 14-Km. 163, 5-34149 Trieste, Italy; 3Department of Physics and Materials, Faculty of Sciences at Cacak, University of Kragujevac, 32000 Cacak, Serbia; jelena.purenovic@ftn.kg.ac.rs; 4Italian Interuniversity Research Center on Interaction between Electromagnetic Fields and Biosystems (ICEmB), Università Degli Studi di Genova, DITEN, Via all’Opera Pia 11/a, 16145 Genova, Italy; 5Interdepartmental Research Centre for Nanomaterials and Nanotechnology at the University of Salerno (NanoMates), Department of Physics, University of Salerno, Via Giovanni Paolo II 132, 84084 Fisciano, Italy

**Keywords:** nanotechnology, functional nanofibers, cardiovascular scaffolds, tissue regeneration

## Abstract

As cardiac diseases, which mostly result in heart failure, are increasing rapidly worldwide, heart transplantation seems the only solution for saving lives. However, this practice is not always possible due to several reasons, such as scarcity of donors, rejection of organs from recipient bodies, or costly medical procedures. In the framework of nanotechnology, nanomaterials greatly contribute to the development of these cardiovascular scaffolds as they provide an easy regeneration of the tissues. Currently, functional nanofibers can be used in the production of stem cells and in the regeneration of cells and tissues. The small size of nanomaterials, however, leads to changes in their chemical and physical characteristics that could alter their interaction and exposure to stem cells with cells and tissues. This article aims to review the naturally occurring biodegradable nanomaterials that are used in cardiovascular tissue engineering for the development of cardiac patches, vessels, and tissues. Moreover, this article also provides an overview of cell sources used for cardiac tissue engineering, explains the anatomy and physiology of the human heart, and explores the regeneration of cardiac cells and the nanofabrication approaches used in cardiac tissue engineering as well as scaffolds.

## 1. Introduction

Cardio Vascular Diseases (CVD) are one of the most prominent causes of mortality in both genders worldwide; thus, they have been marked as a significant public health burden. According to the World Health Organization’s (WHO) estimation, in 2019, 8.9 million people died because of cardiovascular diseases, accounting for 32% of all deaths worldwide. CVDs are linked to heart and vascular problems, including the most frequent atherosclerosis, cerebrovascular disease, ischemic heart disease, and strokes. Clinically, these diseases occur in the mid or older age, after years of unhealthy diet habits, lack of physical activity, alcoholism, and smoking, while obesity, high levels of cholesterol, blood pressure, and diabetes are common risk factors [1].

The apparent global rise in heart failure prevalence is not always related to an increase in heart failure incidence, which, indeed, has been found to be constant or even declining in some studies [2]. The population’s aging, combined with improved heart failure survival (as a result of advancements in treatments and diagnostic technology), may account for the increase in prevalence, while the decrease in incidence (thanks to prevention programs) may be attributed to the treatment of acute coronary syndromes. Among them, cardiac transplantation remains the most effective and efficient treatment. However, the persistent shortage of donor organs and tissues is a serious obstacle to transplantation. Additionally, heart transplant recipients confront severe obstacles to their long-term survival in the form of acute immunosuppression and chronic immune refusal. In light of these observations, there is, therefore, a strong need to find new ways to improve heart failure care [3].

All the issues related to cardiac transplantation and the limited capacity of cardiomyocytes to repair the heart after an acute myocardial infarction (MI) are the primary reasons why much research in regenerative medicine has been developing.

Tissue engineering is a technique that integrates biological sciences and engineering to create functional tissue analogs, such as synthetic heart tissue, which may be used in place of an organ or as mechanical aid [1].

The successful production of contractile cardiomyocytes (CM) from human-induced pluripotent stem cells (hiPSCs) [4] is a significant development in cardiovascular tissue engineering [5]. This is because CMs are normally non-proliferative in the myocardium. Some studies have shown that hiPSC-CM-derived synthetic myocardial tissue is effective in treating heart failure in both small and large preclinical animal models. Some obstacles exist that prevent the widespread clinical use of hiPSC-CM-derived engineered cardiac tissue [6,7,8,9,10]. Scalable three-dimensional (3D) created tissues face challenges in several areas, including survival, electrical integration, immunological response, and maturity/function of hiPSC-CMs (Figure 1).

Two primary technologies are helpful in the fields of tissue regeneration and tissue engineering: (1) biomaterial technology for the creation of three-dimensional porous scaffolds to facilitate direct tissue formation from dissociated cells; and (2) bioreactor cultivation of three-dimensional cell constructs during ex-vivo tissue engineering that aims to recreate the normal stresses as well as flows experienced by heart tissue [11]. Both of these technologies are currently in development. Scaffolds, which have a porous structure that allows for the diffusion of donor cells and growth factors to stimulate and direct the growth of new, healthy tissue; and hydrogels, which are water-insoluble, cross-linked polymer matrices with a high water content that are easily able to be infused into damaged heart tissue and fabricated into a broad range of tissue engineering constructs. Scaffolds and hydrogels are the two main classes of materials that have been traditionally used in cardiac tissue engineering [12].

The properties of biodegradable polymers, such as their optimal porosity, flexible degradation rates, biocompatibility, as well as elastomeric characteristics (which can mechanically favor tissue contraction, which is an inherent part of cardiac function), have attracted a lot of attention for the applications of cardiac tissue engineering. These characteristics include the capacity of scaffolds to maintain their mechanical properties during the process of tissue growth, their capacity to gradually degrade into biocompatible products, and their capacity to accept cells, growth factors, and other such components [13,14].

The two principal classes of biodegradable polymers are synthetic and natural polymers. Natural biodegradable polymers (NAbioPOLY) are polymers derived from the environment. Collagen, gelatin, fibrin, alginate, chitosan, matrigel, and silk are examples of natural polymers. Moreover, other natural nanofibers are also present, such as mussel-derived silk, spider-based nanofibers, sea-silk-based nanofibers, and snail-based nanofibers. These natural nanofibers, along with silk nanofibers created from silkworms, have been utilized in a variety of tissue engineering applications. Natural biodegradable polymers have a number of advantages, including abundant accessibility, biodegradability, and renewability, while their disadvantages include insufficient electrical conductivity, rapid degradation, weak mechanical properties, and immunoreactivity. Man-made polymers are referred to as synthetic polymers. These involve Poly(lactic acid), Poly(glycolic acid), polycaprolactone, Poly(glycolic acid), polycaprolactone, etc.

In addition to controlled structures, stable mechanical properties, flexibility, and the absence of immunological concerns, synthetic biodegradable polymers also exhibit lower biocompatibility and lack of cell attachment. Researchers have developed novel natural/synthetic composites in an effort to mitigate the drawbacks of both natural and synthetic polymers (a combination of both natural and synthetic polymers). In this manner, composite properties have been significantly enhanced. Included in this category are PLA/chitosan, TiO2-PEG/chitosan, and gelatin/PCL/graphene. Natural/synthetic composites exhibit superior biocompatibility, mechanical strength, electrical conductivity, and biological properties [15].

In this review, Natural biodegradable polymers (NAbioPOLY) for cardiac tissue engineering are specifically focused (Section 3) after an overview of the perspective on cardiac tissue engineering (Section 2), addressing cell sources for cardiac tissue engineering, anatomy and physiology of human heart, regeneration of cardiac cells, and technology processes such as nanofabrication, scaffolds, and 3D Bioprinting.

## 2. Perspective on Cardiac Tissue Engineering

### 2.1. Cell Sources for Cardiac Tissue Engineering

Easy harvesting, proliferative, nonimmunogenic, and the ability to develop into mature, functioning cardiomyocytes are all necessary for the ideal cell source to produce an engineered myocardial patch. Even though donor (allogenic) cells are more accessible, they pose a danger of immunosuppression. In contrast, autologous cells have no immunologic obstacles but are more difficult to collect and grow [16]. What follows is the list of potential cell sources for cardiac tissue engineering:Fetal Cardiomyocytes [17,18]Skeletal Myoblasts [19,20]Mesenchymal Stem Cells [21]Smooth Muscle Cells [22]Endothelial Progenitor Cells [23]Crude Bone Marrow [24]Umbilical Cord Cells [25]Fibroblasts [20,26]Human Embryonic Stem Cells [27]Cloned Cells [28]

Myoblasts, which are generated from skeletal muscle, as well as crude bone marrow mononuclear cells, are now the most popular cell types employed for cardiac cell therapy in human patients [29]. Both have benefits over other hypothesized cell types for cardiac repair due to their accessibility, autologous nature, and ease of expansion in vitro. Regarding myoblasts, their apparent inability to transdifferentiate into cardiac or endothelial cells is a drawback to their utility. On the contrary, bone marrow-derived stem cells are currently trending upward in popularity due to their apparent plasticity, which may allow them to modify their phenotype in response to inputs from the target organ. Recently, clinical trials have supported the simple extemporaneous reinjection of unfractionated bone marrow cells in patients with acute MI [29]. However, the applicability of these findings to persistent infarcted myocardial is questionable because they were conducted soon after the ischemic insult [16].

### 2.2. Anatomy and Physiology of Human Heart

The human heart is an organ that pumps and delivers oxygenated blood throughout the body, characterized by four chambers. It is about the size and shape of a man’s clenched fist. Oxygenated blood is exchanged between the lungs and the blood that has returned to the body. Once oxygenated, the blood is disseminated throughout the whole body. Histological and cytological analyses of the heart are required for diagnostics, and this may be useful for understanding rejection in a cardiac transplant or for imitating the organ for tissue engineering.

The posterior part of the heart, which is also known as the base, has a highly compact structure known as the fibrous or cardiac skeleton. The functionality of the skeleton includes an anchoring valve as a robust frame for the cardiomyocytes and as electrical insulation separating atria and ventricles from conductivity. Heart mechanical vibrations coupled with the electric counterpart have been analyzed in order to investigate both the biochemical and the biophysical mechanisms underlying the relationship between heart and brain activity [30]. The human heart is walled in a pericardial sac lined with parietal layers. The wall has three layers: epicardium, myocardium, and endocardium. The innermost visceral pericardium and outermost parietal pericardium are the two pericardial layers. The epicardium is home to both the visceral pericardium, adipose, and fibro/elastic connective tissues. The coronary arteries, veins, lymphatic vessels, and nerves pass under the epicardium. Endothelium and sub-endothelial connective tissues constitute the whole endocardium, and it is an important organ. The impulse conducting system is situated in the sub-endocardium [31].

The heart’s impulse-conducting system is comprised of cardiac cells that are specially designed to conduct electrical impulses. Human cardiac muscle is made of a variety of cell types, with cardiomyocytes (CMs), pericytes, endothelial cells, and fibroblasts being the most prevalent. Although the percentage of each cell type in the heart is still debated, there is evidence that CMs occupy the bulk of the heart volume (about 75%) and comprise 30–40% of all cells. Electrical impulses are produced at the site of the sinoatrial (SA) node, located at the junction of the right atrium and superior vena cava. Between the interatrial and interventricular septum, these impulses flow through the atria to the atrioventricular (AV) node. After entering the fibrous central body of the cardiac skeleton, the fibers migrate inferiorly and pierce it. In the interventricular septum, these fibers are Purkinje fibers, which branch into the ventricles after dividing [32]).

### 2.3. Regeneration of Cardiac Cells

Tissue regeneration requires a process called mitosis, where cells divide and proliferate. Initially, it was assumed that heart muscle cells, or cardiomyocytes, were incapable of regeneration and that their total number was fixed at birth. During the last two decades, researchers have found that these cardiac cells have reduced proliferative activity, but it is still insufficient to restore functionality after ischemic or other cardiac injuries. The reduction in cell division or regeneration results in the loss of Lamin B2, which causes the cells to adhere together when they divide, resulting in the presence of more than two copies of the organism’s genetic code (chromosomes) in each nucleus. Once these additional chromosomes are present, cell division becomes more complicated [33].

There are several critical mechanisms underlying cardiac tissues repair and their regeneration, shown in Figure 2, including (1) protection and its survival, (2) reduction in inflammation, (3) cell-to-cell communication, (4) angiogenesis or vascularization, (5) cardio myogenesis, (6) regulation of cell cycle and proliferation at the molecular level, and (7) cardiac tissues aging linked with weakened regenerative and reparative potential [34].

At least two methods are effective in regenerating the heart. To begin, cardiomyocytes may possibly be produced from induced pluripotent stem cells or embryonic cells and injected into the heart either as cell suspensions or as contractile myocardial tissue formed ex vivo. Alternatively, the endogenous proliferation capacity of cardiomyocytes may be increased via the introduction of certain genes or, more effectively, chosen microRNAs. The idea that cardiac muscle regeneration may be accomplished either via the implantation of ex vivo-produced CM or by stimulating the proliferative potential of endogenous cells remains intriguing. Thus, currently, stem cells and endogenous regeneration are still far from being widely accepted in the therapeutic realm. Cell-derived cardiac mesenchymal precursor cells (CM MP) are young, and the electrical and mechanical connection to their endogenous counterparts is restricted. Regardless, the procedure of producing large numbers of CM MP for each patient is complicated and expensive to scale up. When using CM technology, the maturity of the material seems to increase, but the challenge of integrating vast patches of engineered myocardium is still unresolved [35].

### 2.4. Nanofabrication Approaches in Cardiac Tissue Engineering

A general overview of the nanofabrication approaches used in cardiac tissue engineering is given here, whereas more accurate and focused information on the application of these methods to obtain Natural biodegradable nanopolymers is provided in a dedicated Section (see Section 3). In 2009, Mironov et al. described biofabrication as the synthesis of complex biological products from raw materials such as living cells, chemicals, extracellular matrix, and biomaterials [36]. Conventional scaffold fabrication methods were commonly used in early tissue engineering studies to produce 3D constructs with manipulatable biological and physicomechanical properties. These methods included lyophilization or freeze drying method, chemical cross-linking, cryo-precipitation, and electrospinning [37]. Many of these techniques are still in use today (electrospinning in particular), but they do not allow for the distribution of cells and biomaterials or the creation of complicated geometries. There are two main types of biofabrication approaches, known as top-down and bottom-up, which are based on the methodological approach taken to the whole process [38].

Traditionally, tissue engineering has been approached from the top down, with cells planted on a biodegradable polymeric scaffold, such as polyglycolic acid (PGA). Top-down methods anticipate cells populating the scaffold and establishing the proper extracellular matrix (ECM) and microarchitecture, frequently with the assistance of perfusion, growth hormones, and/or mechanical stimulation. Despite advancements in surface patterning and the use of more biomimetic scaffolding, such as decellularized ECM copies, top-down methods frequently struggle to replicate the complex microstructures of tissues [39]. Recently, the biologically inspired modular tissue engineering technique has emerged as a potential fabrication strategy to solve the common drawbacks of a prefabricated scaffold through the assembly of tissue constructions from the bottom up [40]. As an alternative to the traditional scaffolds’ top-down method, this theory allows for the efficient engineering of complex tissues and organs from a microscale module. This strategy is gaining traction as a useful tool for studying and modeling vascular physiology in tissue engineering [41,42]. 3D tissue printing [43], cell sheet technology [44], and the assembly of cell-laden hydrogels [45] are just a few examples of the modular TE solutions proposed so far (Figure 3).

The two primary methods for cardiovascular tissue engineering presently being used are scaffolds or cardiac patches and in-situ gelling devices. Scaffolds or cardiac patches are porous and dense polymeric matrices that might include cells or bioactive chemicals. These methods are restricted by the intrusive process required for the implantation, which is not an alternative for patients with advanced heart failure that are unable to undergo surgery due to comorbidities. In the 1990s, ventricular constraint techniques like polymeric meshes encircled the heart or stitched it back to its surface. A number of studies have demonstrated that they are effective at reducing infarct expansion in large animal models by mechanical stabilization of the heart and restricted long-term changes in the left ventricular geometry, but with limited clinical translation options, cardiac patches and in situ gelling systems are the two primary approaches being investigated for cardiac tissue engineering at the moment [46].

Hydrogels have garnered significant interest in the gel category as the “water-swollen polymer networks” with higher water content comparable to living tissue. They may be made of synthetic or natural polymers and can absorb a significant quantity of water or biofluids while maintaining their form. As injectable fluids, they enable the targeted administration of bioactive compounds to specific locations while maintaining the kinetics and extending carrier molecules’ functional half-life in a well-controlled manner. Hydrogels may also be utilized as a delivery medium for “combination methods”, encapsulating cells and bioactive compounds concurrently [47].

#### 2.4.1. Lyophilization or Freeze-Drying Technique

The method of lyophilization, also known as freeze-drying, is essential and well-established for increasing the long-term stability of unstable pharmaceuticals, particularly therapeutic proteins [48]. Approximately half of all biopharmaceuticals on the market today are lyophilized, making it the most popular method of formulation [49]. Long-term stability is enhanced in the freeze-dried solid form because physical or chemical degradation reactions are either prevented or significantly slowed down [50]. Lyophilized formulations have improved stability and are simple to handle, ship, and store [48].

Freezing, primary drying, as well as secondary drying make up the three phases of the standard lyophilization cycle. The liquid formulation is chilled till ice nucleates, and then ice grows during the freezing process. Most of the water is precipitated out as ice crystals, leaving behind a glassy and/or crystalline matrix of solutes [51,52]. The crystalline ice that was produced during freezing is sublimated away during the first drying stage. To compensate for the heat lost through the sublimation of ice, the chamber pressure is lowered below its vapor pressure, and the shelf temperature is increased. When primary drying is complete, the product may still retain 15–20% of unfrozen water; this water is desorbed during secondary drying, which typically takes place at high temperature and low pressure [53].

Lyophilization, in general, is a drying process that requires a lot of time and energy. The freezing process usually lasts only a few hours, but the drying can take several days. When comparing the drying process itself, secondary drying takes only a few hours, while main drying can take several days. For this reason, researchers have focused on finding ways to minimize the primary drying time without compromising product quality during the development of lyophilization cycles [54]. Despite its key role in the lyophilization process, freezing has been largely overlooked in the past [55].

#### 2.4.2. Chemical Cross-Linking

Hydrogels are water-loving polymers that are held together by physical or chemical cross-linking and hence do not dissolve [56]. Hydrogels’ biochemical and mechanical properties are strongly linked to the cross-linking techniques used to create them, and even hydrogels made from the same components but with various cross-linking structures can exhibit distinct behaviors [57].

The cross-linking generation of stable polymeric networks is the primary fabrication method for hydrogels. Chemical and physical crosslinks are two fundamental ways among many cross-linking mechanisms [58,59,60]. Thermal induction based on LCST (Lower Critical Solution Temperature) [61,62,63], UCST (Upper Critical Solution Temperature) [64,65] and ultrasonication [66,67,68] mediated sol-to-gel phase transition creates physical crosslinks through hydrogen bonding, ionic/electrostatic interaction, crystallization/stereo-complex and hydrophobic interactions. Photo-polymerization, enzyme-induced crosslink, and “click” chemistry (Michael type-addition, Diels–Alder “click” reaction, oxime production, Schiff base formation, and so on) are all examples of chemical cross-linking [61,69,70,71]. These cross-linking techniques may motivate us to develop new hydrogels with improved architectures and desirable characteristics.

#### 2.4.3. Cryo-Precipitation

As a biofabrication method, cryo-precipitation uses low temperatures to produce substances with novel chemical and physical features. A liquid solution is cooled down below the freezing point in this method. Some of the solution’s ingredients become insoluble at this temperature, settling out as a solid precipitate. Once the precipitate has formed, it can be molded into a wide variety of forms. The fields of biomedical engineering, tissue engineering, and drug delivery can all benefit from cryo-precipitation. Biocompatible scaffolds for tissue engineering can be made from the resulting nanofibers. Cryo-precipitation has also been utilized to make biodegradable nanofibers for drug delivery systems, which can release medications slowly and steadily over time [72].

#### 2.4.4. Electrospinning

In order to mass-produce nanofibers, electrospinning has traditionally been utilized [73] because it can produce nanofibers with a wide range of morphologies. Nanofiber matrices may be efficiently synthesized using electrospinning. Electrospinning is a highly adaptable procedure, and its matrices have many applications. Some of these include air and water treatment filtration processes, biosensors, affinity membranes, cell regeneration, solar cells, cosmetics, drug delivery systems, textiles, tissue engineering, and wound dressing [74].

In order to conduct electrospinning, three primary elements are required: a metallic needle (spinneret), a high-voltage power supply, and a collection plate permanently linked to the spinneret (a grounded collector). Initially, the polymer solution is stored in a syringe as part of its mechanism. Accuracy requires that the polymer solution in the syringe be completely devoid of air bubbles. A metal needle is attached to the syringe. When supplying the metallic needle with solution, a syringe pump controls the rate of flow. To make a “Taylor cone” [75], a high-voltage power supply (often in the range of 1–30 kV) is delivered to a droplet of the polymer solution at the nozzle of the metallic needle. Induced charges are spread out evenly throughout the droplet’s surface, and it becomes electrified and stretched. The droplet encounters two primary forms of electrostatic forces: electrostatic repulsion and Coulomb forces. An external electric field exerts Coulomb forces, whereas electrostatic repulsion balances surface tension. To create consistent and continuous nanofibers, the electrified jet is stretched and thrashed before being deposited on the counter-electrode [76]. These homogeneous nanofibers with sizes on the nanometer scale are captured on a collection plate. The polymer solution is evaporated from the needle onto the collecting plate. The nanofibers will settle on the collector, which may then be used to collect them. Rodoplu and Mutlu (2012) [77] describe two types of nanofiber collectors, horizontal and vertical, distinguished by the orientation of the discharging capillary and the collecting target. Plate collectors come in a variety of shapes and sizes; some of the most common include the cylinder and disc plate collectors mentioned by Shahriar et al. (2019), the horizontal and vertical plate collectors mentioned by Cavo et al. (2020), and the disc, grid, liquid bath, mesh, parallel bar, pin, rotating drum with wire wound on it, rotating cylinder, and rotating rods mentioned by Bhattarai et al. (2018) [73,78,79,80,81]. Figure 4 depicts the mechanics of the electrospinning method used to create nanofibers.

#### 2.4.5. Bubble Electrospinning

Bubble electrospinning, introduced in He and Liu’s 2012 seminal paper [82], is a technique that has recently joined the ranks of electrospinning’s other cutting-edge technologies. When it comes to the electrospinning manufacturing of submicron fibers, bubble electrospinning offers many benefits. It is used to create bubbles by means of electric forces to break the surface tension. Forced air causes these bubbles to form on the surface of the polymer solution. Taylor cones, formed by electric forces, give rise to the fibers. The surface tension is proportional to the bubbles’ size and shape. The Young–Laplace equation can be used to describe the bubbles’ surface tension: The surface tension is given by: $\sigma =\frac{r\Delta P}{4}$, where r is the bubble’s radius and ΔP is the pressure gradient [83].

Some uncontrollable bubbles form on the surface of the solution during the electrospinning process. However, the fiber deposition features, morphology, and output of the industrial production process are unaffected by this phenomenon [83].

The idea of electrospinning as a technique inspired the development of the first laboratory electrospinning apparatus, which could be updated or used as the foundation for a brand new type of bubble electrospinning apparatus [84]. Possible medical uses include the development of novel composite nanofibrous scaffolds in tissue engineering [85]. In-depth information about the machinery used in bubble electrospinning to create fibers is provided in this study. The described experiment evaluates the efficiency and output structures of AC bubble electrospinning technology in comparison to the more conventional DC bubble as well as needle electrospinning methods [86].

### 2.5. Scaffolds for Cardiovascular Tissue Engineering

For efficient tissue engineering, the scaffold is one of the most significant factors to think about because its surface properties, interface adherence, external geometry, pore density and size, biocompatibility, mechanical properties, and degradation affect not only the creation of the tissue construct in vitro but also its viability and functionality after implantation [87,88]. Basic parameters such as biocompatibility, sterilizability, and mechanical integrity must be met by any scaffold intended for use in tissue engineering. Direct contact with blood presents extra unique obstacles for scaffolds used for heart valve tissue creation. In particular, the structure must be able to withstand calcification and thrombosis [89]. Furthermore, from the time of implantation onward, the construct must be able to tolerate the specific hemodynamic pressures and flows of the cardiac environment. Because of these specific difficulties, it is crucial to give considerable thought to the scaffold’s materials and architecture before making a tissue-engineered heart valve [90].

Human pulmonic and aortic semilunar valves have three semicircular leaflets (also known as cusps) connected to the root, a fibrous annulus. All three of the leaflets’ layers—the fibers (fibrosa), glial cells (spongiosa), and blood vessels (ventricularis) are reported in Figure 5). Layers of valvular interstitial cells (VICs) are embedded in a matrix of collagen, elastin, and glycosaminoglycans (GAGs). Typical leaf surfaces are almost completely devoid of blood vessels, meaning they rely on hydrodynamic diffusion and convection to draw nutrients and oxygen from the bloodstream. The aortic or pulmonary root, on the other hand, is a fibrous structure in the shape of a bulb with three layers: the intima, the media, and the adventitia. Specifically, endothelial cells populate the intima, smooth muscle cells the media, and fibroblasts the adventitia. In a nutshell, the geometry of semilunar heart valves is intricate. To create engineered heart valves with a similar structure, researchers must first develop a scaffold with the right surface properties, morphologies, mechanical properties, and pore diameters. Understanding the whole range of available tissue engineering scaffolds for heart valve repair is crucial in this regard [91].

### 2.6. 3D Bioprinting Technology for Cardiac Tissue Engineering

The term “3D bioprinting” refers to a newly emerging category of manufacturing technology that can create three-dimensional (3D) constructs with bio-activities or bio-functions, typically through a layer-by-layer sequence of complex depositions based on a digital pattern [92,93]. Various fixed systems, including needle-droplet printing, pneumatic extrusion printing, and photocuring-based printing, have been derived from the reconstruction techniques and associated functions relating to the 3D digital standardization of source materials as the additive manufacturing industry has progressed [94,95]. For extrusion and droplet bioprinting, the materials that are available per unit of time mostly rely on external mechanical force as well as gravity to build a 3D structure along a set path. On the other hand, laser-assisted bioprinting uses sensitive optical guidance as the primary way to shape materials. Most of the time, what these methods have in common is that they all use coagulative rheological “inks” for quick volumetric prototyping [96].

Implementations of 3D bioprinting technology in the area of cardiovascular repair and regeneration have made significant progress over the past decade, especially those aiming to create holistic engineered heart tissue (HEHT) with prolonged contractility for transplantation. Importantly, the prototypically established route for the fabrication of HEHT using a 3D bioprinting system, like the dominant extrusion-based 3D bioprinter [97], is heavily dependent on the precise depositions of cardiac-related cell-laden polymerous bioink in an orderly manner. With the advancement of micro-scale control in 3D printing technology and the cardiac-focused advancements of iPSCs and biomaterials, the realistic layout of this mechanical operation has gradually deepened [98]. In spite of this, the new area that has arisen as a result of the overlap between these disciplines is bursting with life and holds great promise for the advancement of cardiology in the clinic in the future [99,100,101]. First-hand experience in vitro has led to a focus on 2D culture systems for cell expansion, while approaches to stem cell differentiation, as well as somatic cell trans-differentiation, have tended to be more exploratory and amenable to optimization [102,103]. The ability to obtain and maintain a stable supply of cardiacmyocytes (CMs) or progenitors suitable for 3D bioprinting a cardiac system is crucial because these cells are the fundamental building block of any structural and functional bio-simulator. Acquisition, cultivation, as well as encapsulation are the three stages of preparation that must precede the formal printing operation with active cardiac-bioink cells. At present, human, mouse, or rat heart cells can be used in clinical applications, and the primary cells (such as primary rat ventricular CMs as well as human coronary artery endothelial cells), cell lines (such as H9C2), or iPSCs, as well as other progenitor cells derived from iPSCs, are all included in the first-hand genre [103,104].

The fundamental step of 3D bioprinting is the preparation of adequate bioink. Mostly, bioinks are solely composed of cells and do not involve any hydrogel component to retain biocompatibility [96]. Once the bioink has been prepared, a series of sequential fabricating operations steered by integrated computer numerical control machinery (CNCM) must be performed [105] in order to create a biosimulated network of cardiac cells and bio-materials for evaluation and/or application [106]. After 3D bioprinting is complete, the engineered structure’s cardiac biological functions must be carefully implemented in vitro and in vivo for extensive follow-up research. Cardiac biomarkers (CBMs), efficient contractile forces (ECFs), spontaneous action potentials (SAPs), and total calcium regulation (OCR) are all evaluable in vitro, while in situ engraftment can help repair and regenerate damaged tissue [107,108].

## 3. Natural Biodegradable Nanopolymers Used in Cardiac Tissue Engineering

Recent advances in the incorporation of nanoparticles into tissue engineering have been made [109]. Nanomaterials may be particularly useful in cardiovascular tissue engineering. The latter is concerned with regenerating damaged cardiac tissue, sometimes through the induction of cell proliferation with regenerative potential, such as mesenchymal stem, embryonic stem, as well as induced pluripotent stem cells [110]. Numerous efforts are being made to develop polymer scaffolds or patches that can be used to support tissue regeneration or repair. Cardiovascular tissue engineering has been tested by using biomaterials such as hydrogels, electrospun polymers, and 3D-printed cardiac patches [111]. Additionally, injectable hydrogels have been developed to facilitate clinical use by allowing for easy delivery to injured myocardium without the need for invasive approaches. Following modification with bioactive molecules such as microRNA, peptides, or growth factors, these materials may be repurposed. Although these biomolecules are susceptible to degradation in a physiological environment, nanoparticles can aid in their stabilization. Nanoparticles have been found to be beneficial in these efforts by stimulating neighboring cells and acting as platforms for bioactive molecule modification [15,112].

Synthetic polymers and natural polymers are the two main types of biodegradable polymers. Polymers that are naturally biodegradable are referred to as NAbioPOLY. Synthetic polymers are those that are created in a laboratory. This category includes substances such as poly(lactic acid), poly(glycolic acid), poly(glycolic acid), poly(caprolactone), etc. [15]. Natural polymers include things such as silk, gelatin, collagen, chitosan, alginate, and chitosan. There are pros and cons to using natural biodegradable polymers. Pros include their abundant availability, biodegradability, and renewability; cons include their low electrical conductivity, quick disintegration, weak mechanical capabilities, and immunogenicity (Figure 6).

Natural polymers are derived from natural sources, for example, plants and animals. These natural polymers are formed of nano-structured molecules and are, therefore, classified as nanomaterials [113]. Because of their biocompatibility, renewability, biodegradability, and widespread availability, these natural polymers have been employed in different cardiovascular tissue engineering applications [114]. NAbioPOLYs such as collagen [115], gelatin [116], fibrin gel [117], alginate [118], chitosan [119], matrigel [120] and silk [121] are frequently employed in cardiac tissue engineering. They will be discussed in the following subsections.

The primary feature of NAbioPOLYs that puts them at the frontline of cardiac tissue engineering is their biocompatibility. Additional important properties that have been studied and found to be acceptable for cardiac applications include mechanical properties and rate of degradation. Hydrogels are soft materials among these biomaterials, and they are ideally suited for cardiac repair because they are made from naturally occurring matrices such as collagen, gelatin, fibrinogen, alginate, matrigel, chitosan, and silk. To date, there have only been a handful of clinical trials using hydrogels made from NAbioPOLYs [122].

In order to more closely resemble normal tissue structure, including cardiac tissue, NAbioPOLYs have been transformed into other types of 3D structures with tailored porosity. These are advantageous because they can be processed using a wide variety of methods and boast a wide range of mechanical properties that can be fine-tuned to meet the needs of individual patients. The development of big perfusable vessels is still difficult, despite the fact that many approaches have been taken to increase vascularity inside the cardiac patches. To avoid the potentially fatal arrhythmic effects of cell injection in large quantities, more study is needed to determine how to enhance functional coupling between the graft and host cardiomyocytes. Intriguing advances in cardiac tissue engineering are on the horizon thanks to multi-material structures made possible by 3D printing techniques and featuring structures tailored to individual patients. These scaffolds aim to mimic the properties of natural cardiac tissue as closely as possible [123].

### 3.1. Collagen

Collagen is the most frequently employed natural polymer in tissue engineering since it is found in the ECM of nearly every human tissue. Its use has increased due to its good biocompatibility and weak antigenicity. Collagen has been utilized in a variety of applications over the years due to its pro-vascularization biocompatibility, high cellular activity, hyposensitivity, biodegradability, and low toxication [124]. Collagen provides several advantages for cardiac tissue engineering, including heat reversibility, biocompatibility and high cellular activity, hyposensitivity, biodegradability, and low toxication [125].

Collagen, comprising skin, bone, tendons, ligaments, and cartilage, is found in almost all human tissues. Collagen types I, II, III, and IV are frequently explored in tissue engineering [126]. Among them, thanks to its biocompatibility, type I collagen is mostly utilized in tissue engineering [127]. It comprises 75–85% [122] of the ECM in the heart and has the added benefit of being relatively nonimmunogenic. Although collagens can be subdivided into fibrillar and non-fibrillar components, type I collagen consists of two alpha-1 chains and one alpha-2 chain, forming long fibers whose characteristics are dependent on the density and spatial alignment [128]. These non-fibrillar parts can bind to membranes or create networks [129,130].

The human body, and especially the native myocardium, are primarily composed of collagen I. The latter, like all collagens, has a three-helical structure at its molecular level. These molecules, when left to their own devices, self-assemble into fibrils, which are then organized into collagen fibers of variable diameters in the body. As a result, the tensile mechanical contributions of these fibrils and fibers are maximized in tissues subjected to high mechanical stresses, such as tendons, ligaments, and muscles. Long channels surrounding cardiac muscle bundles and bestowing to the anisotropy of native myocardium are formed by collagen fibers, the primary component of the endomysium in the myocardium. It has been shown that resident cells can apply stresses to initially disordered collagen hydrogel scaffolds, causing them to become organized over time [131,132].

Collagen-based biomaterials have been the focus of recent studies for the treatment of illnesses such as myocardial infarction. Particles, like growth factors or peptides, can be transported by these materials to stimulate differentiation as well as patterning [133]. Based on these preliminary findings, researchers have intramyocardial injection methods, as they allow for targeted, direct administration to the heart muscle. However, surgery is required for this method, and there is also the risk of the substance leaking out into the surrounding tissue [134]. So-called “cardiac patches” are a substitute for this kind of administration. Cardiac patches contain certain qualities that are not unique to collagen but are nonetheless useful. These patches have the ability to infiltrate models and have high engraftment levels [135], and one feature is the possibility of cultivating cells ex vivo to encourage the proper invasion of the patch. Eventually, autologous bone marrow mononuclear cells seeded into a 3D collagen type I matrix (for the regeneration of ischemic myocardium) increased the thickness of the infarct scar with viable tissue, assisted normalized cardiac wall stress in wounded regions, limited ventricular remodeling, and improved diastolic function [136] (Table 1).

For cardiac tissue engineering, scaffolds of collagens, particularly nano-fibrous scaffolds, are being studied. Punnoose et al. (2015) describe one of the most straightforward new approaches for the development of nano-fibrous scaffolds from collagen by the use of an electric field among the solution of polymer and grounded collector. Fluoroalcohols (e.g., 1,1,1,3,3,3–hexafluoro-2-propanol (HFIP) and 2,2,2-trifluoroethanol (TFE)) to manufacture nano-fibrous scaffolds from collagen type I, which generally has been the favored for the manufacture of biomaterials based on collagen. Though fluoroalcohols are caustic and expensive, numerous investigations have been conducted with the goal of discovering a benign and economically viable solvent. For instance, Punnoose et al. (2015) stated the utilization of a simple benign binary solvent solution comprised of acetic acid and dimethylsulphoxide for electrospinning the nanofibers from type I collagen whose diameter ranges from 200–1100 nm in one study. This solvent was not just inexpensive but also preserved the inherent properties of electrospun collagen [137].

**Table 1 nanomaterials-13-01374-t001:** Collagen scaffolds for heart tissue engineering.

Scaffold	Method of Biofabrication	Evaluated Properties	Source
Collagen/graphene oxide cardiac patch	Freeze-drying Method	Connected pores of the right size and electrical conductivity that are just right for use in cardiac tissue engineering; non-toxic resultant changes in human cells, cardiomyocyte adhesion in neonates, and expression of cardiac genes.	[138]
Injectable hydrogel (Collagen/carbon nano tubes/chitosan/gold nanoparticles)	Chemical Cross-linking	Non-toxic, promising heart tissue engineering biomaterial.	[139]
Collagen/chitosan composite scaffold	Freezing and lyophilization	Biocompatibility, strong expression of cardiac-specific marker protein, contractile performance, high porosity (>65%), and mechanical qualities in the physiological range of native myocardium are all hallmarks of CM.	[140]
Conductive nanofiber scaffold (polypyrrole hydrogel/chitosan/polyethylene oxide)	Electrospinning	Cell adhesion, proliferation and growth, nanofiber scaffolds appropriate for use in internal organs with electrical impulses like cardiovascular tissue engineering.	[141]

### 3.2. Gelatin

Gelatin is a natural polymer that is similar to collagen, from which it is derived. Once collagen has been separated out, gelatin can be extracted in one of two ways: alkaline hydrolysis or acid hydrolysis. The latter is how the IP (Isoelectric Point) of gelatin is calculated. Type A, with an IP value less than 5, is the designation given to gelatin after it has been acid hydrolyzed. Type B gelatin, with an IP of less than 9, is the result of extraction in an alkaline medium. Denaturalization gives gelatin its characteristic linear structure, made up of Gly-X-Y (mostly proline and hydroxyproline) sequences. The RGD (Arginylglycylaspartic acid) motif, a different set of amino acids in the structure, also aids in cell adhesion, proliferation, and differentiation [142,143,144,145].

In addition to promoting cell adhesion, differentiation, and proliferation without eliciting an immune response, its biocompatible, biodegradable, and low-toxic properties also make it easily digested by the body’s own enzymes (metalloproteinases) [146,147]. Because of its low price, it has been used in a variety of applications (microparticles for bone regeneration enhancement, wound dressing, hydrogels for the controlled release of chemotherapeutic agents in the treatment of cancer) and in a wide variety of tissues (bone, skeletal, neural) [148,149].

However, gelatin is renowned for its capacity to take in liquid. Porosity guarantees a diffusion of nutrients and oxygen for proper cell growth, making it a highly desirable property in tissue regeneration [150]. Nevertheless, some gelatin-based cell delivery systems have evidenced a poor cell survival rate [151], suggesting that porous structures do not always fulfill every requirement for facilitating the exchange of products for cell survival. Finding a way to use gelatin-based systems to ensure proper pore size, which may lead to an elevated rate of cell survival, is thus the current research focus [145].

Gelatin is a possible biomaterial for cardiovascular tissue regeneration since collagen is abundantly found in the extracellular matrix of numerous organs, including the heart. Gelatin hydrogels have several limitations, including a lack of durability, mechanical stability, and high-water content. Thus, gelatin-based biomaterials have been produced by cross-linking by enzymatic, chemical, or physical cross-linking. Synthesizing gelatin-based nanofibrous scaffolds is a technique to increase their strength in cardiovascular tissue applications (Table 2). Elamparithi et al. (2016) [152] recently electrospun gelatin nanofibrous matrices and investigated them for primary cardiomyocyte development and function, using a benign binary solvent (acetic acid, dimethylsulfoxide (DMSO)). Channels and grooves bio-printed in three dimensions can have a significant effect on cell behavior, phenotypic, and morphology. Tijore et al. (2018) studied the development of a stem cell myocardial lineage using a gelatin scaffold. They discovered that by the 3D printing hydrogel of gelatin which cross-linked with the enzyme MTGase and reliable cell alignment, micro-channels could be created [153].

### 3.3. Fibrin Gel

Fibrin is a biopolymer that forms naturally during the coagulation process, making it a biomaterial with several potential applications [122,156]. Fibrin is an extensively employed natural polymer in cardiovascular tissue engineering, most notably for the encapsulation of cardiac cells [157]. The fibrin is formed during hemostatic coagulation by the prompt polymerization of fibrinogen monomers at room temperature with the help of the proteolytic enzyme thrombin as a cross-linking agent. The mechanical characteristics and the rates of gelation of fibrin have been shown to be directly connected with the variation of ratio for fibrinogen/thrombin. Due to its better biocompatibility, predictable rate of degradation, natural hydrogel properties, and absence of toxicity, fibrin remained widely promoted and utilized in cardiovascular tissue engineering. This has allowed for the creation of small-diameter vessels that are both resistant to systolic pressures in vivo and immunologically compatible with the patient, thereby reducing the likelihood of graft rejection. New technologies and the adaptability of fibrin (which can be used as glue or as engineered microbeads) have increased the biopolymer’s utility. Adipose, bone, cardiac, cartilage, liver, nervous, ocular, skin, tendons, and ligaments are just some of the tissues that have been successfully regrown using fibrin as a biological scaffold alone or in combination with other materials. Bioactive peptides, as well as growth factors delivered via a heparin-binding delivery system, can further enhance its efficacy. The geometry of its structure can be changed into appropriate and predictable forms using cutting-edge technologies such as inkjet printing and magnetically influenced self-assembly. Because of its versatility and adaptability to in vitro manipulation, fibrin provides unique biomaterial properties. In order to create a 3D bioengineered tissue that can be used as an in vitro model system, fibrin gel casting was employed to bioengineer functional vascular smooth muscle (VSM) strips from primary human aortic VSM cells. Microthreads made of fibrin, with a diameter of about 155–165 (μm), can be used to promote the growth and alignment of cells in a longitudinal direction using contact guidance. Compared to simple hydrogels, which also promote cell survival and engraftment, this type of scaffold appears to offer an advantage by more closely approximating the architecture of the target tissue and facilitating cell alignment. Large wounds can benefit greatly from the use of fibrin micro threads, as they eliminate the need for prevascularization [158] (Table 3).

However, it has several important disadvantages, including poor mechanical qualities, gel shrinkage, and the possibility of disease transmission; all of these factors prevent fibrin gel from being an excellent option for tissue engineering [159].

**Table 3 nanomaterials-13-01374-t003:** Fibrin-based scaffolds for heart tissue engineering.

Scaffold	Method of Biofabrication	Evaluated Properties	Source
Fibrin-based cardiac patch	Chemical Cross-linking	The composite patch was biocompatible and even stimulated cardiomyocyte proliferation in vitro thanks to its interaction with NRG-1/ErbB signaling.	[160]
Bio-hemocompatible cardiovascular patches	Cryo-precipitation	Constructing durable and biocompatible circulatory patches with regenerative potential by combining compacted fibrin matrices and spider silk cocoons may be a workable notion.	[161]

### 3.4. Alginate

Alginates are a class of naturally occurring polysaccharides that are considered biodegradable, biocompatible, non-toxic, as well as non-immunogenic [162]. Natural alginate is a polysaccharide that makes up 18–40% of the biomass of some brown algae (including Ascophyllum nodosum, Macrocystis pyrifera, Laminaria hyperborea, and others) [163]. Numerous approaches have been employed to modify the gelation and mechanical characteristics of alginate, including conjugation of several materials, cross-linking, and immobilization of particular ligands (Table 4). Sondermeijer et al. (2017) synthesized a scaffold made of alginate with a covalently bonded synthetic cyclic RGDfK (ArgGlyAspDPheLys) peptide to enhance cell survival and angiogenesis in injured myocardial tissue. The modified version of scaffolds supplied through (Human Mesenchymal Precursor cells) hMPCs was investigated in rats with myocardial infarction, and increased vascularization at the infarct border zone was obtained with scaffolds seeded with 1106 hMPCs compared to scaffolds planted with 3106 hMPCs and scaffolds without cells (7 days). Additionally, one week after implantation, the epicardial scaffolds demonstrated no foreign body response to the scaffold material. As a result, a 3D alginate scaffold purified and enhanced with a cyclic RGDfK peptide was able to boost cell survival and angiogenesis, indicating its non-immunogenic and biocompatible qualities [164].

Bioengineered cardiac grafts, stem cell delivery, acellular injectable tissue support and reconstruction, and the description of multiple growth factors in a manner reminiscent of nature are just some of the applications of alginate hydrogels. Clinical trials of two injectable alginate implants have already begun, demonstrating the good prospect of alginate-based approaches to myocardial repair and regeneration [163].

Cardiac tissue engineering and regeneration are two areas where alginate has shown great potential and versatility. Biocompatibility, mild gelation conditions, and easy improvements to create alginate derivatives with unique properties are the most appealing aspects of alginate for these applications. Several applications have made use of alginate biomaterials, including stem cell delivery, 3D microenvironment design for functional cardiac tissue formation, and bio-inspired design of systems for controlled release and presentation of multiple combinations of bioactive molecules and regenerative factors [163].

### 3.5. Chitosan

Chitosan is a linear polysaccharide comprised of -(14)-linked D-glucosamine and N-acetyl-D-glucosamine. It is formed through the deacetylation of chitin. The chitin deacetylation process not only regulates the degree of deacetylation (DD) but also modifies the average molecular weight of chitosan [168]. Chitosan has low toxicity and good biocompatibility due to its structural resemblance to natural glycosaminoglycans. Chitosan biodegrades into harmless compounds during in vivo tissue applications via enzymatic hydrolysis. Chitosan and its derivatives are frequently used to coat or graft onto scaffold surfaces to enhance cell identification and cytocompatibility in tissue engineering applications [169]. Unfortunately, chitosan is insoluble at physiological pH. Hydrogels based on chitosan have recently been evaluated as a biodegradable material for applications in cardiovascular tissue engineering due to their controllable biodegradability and biocompatible degradation products (Table 5). According to Xu et al. (2017), the temperature-responsive chitosan-based hydrogel was used to transport MSCs. Researchers found that a temperature-responsive chitosan hydrogel has improved cell retention and graft size in an ischemic heart, enhanced the effects of MSCs on the formation of neo vasculature, and promoted cardiomyocyte differentiation in the MSCs. Additionally, this hydrogel has improved the function of the heart and hemodynamics in rats with myocardial infarctions after the five weeks of transplantation in the infarcted area [170,171].

### 3.6. Matrigel

Engelbreth–Holm-Swarm (EHS) murine sarcoma cells produce matrigel, a gelatinous extracellular matrix (ECM) protein that is frequently utilized as a cell culture matrix. Although matrigel is often utilized as a coating for the substrate to promote cell adhesion, matrigel hydrogels are also used to repair heart tissue. Hydrogels composed of matrigel include important cytokines growth factors for the growth of cells and have been used as a cell-compatible gel in the past [175] (Table 6). However, the fact that it is derived from animals makes it unfit for therapeutic use (derived from murine sarcoma cells) [119], Zhang et al. (2017) created the vascularized pacemaker tissues in matrigel using endothelial progenitor cells (EPCs) and cardiac progenitor cells (CPCs). Cardiac sinus node dysfunction was explored by implanting a tissue-engineered cardiac pacemaker (TECP) made from CPC-derived pace-making cells in culture for 21 days and then implanting it in vivo for four weeks. The vascularization of TECP (vTECP) was performed using a mix of CPCs and EPCs (implanted into rat hearts). Individual survival was increased with TECP implantation in sinus node injury animals, and the inserted TECP demonstrated electrical activity. Additionally, the optimal ratio of EPCs to CPC-derived pacemaker cells (1:1) resulted in the greatest vascularization in vitro for vTECP. Additionally, vTECP insertion boosted electrical activity in vivo. Nevertheless, the primary disadvantage was an inefficient vasculature in engineered tissues, which poses a significant barrier to their utilization in cardiac tissue engineering [176].

### 3.7. Silk

The glands of arthropods such as spiders, silkworms, flies, scorpions, mites, and bees are responsible for producing the silky fibers that are commonly referred to as “silks” [178]. Silks from different species have vastly different structures, compositions, and properties. Silkworms are the primary source of the most well-known and widely used silks. If a silkworm does not eat mulberry leaves, it is not considered a mulberry silkworm, and vice versa. Silk from the Saturniidae family, including the Antheraea assamensis and Antheraea mylitta species, is used to make things other than mulberry silk, which is spun by the Bombyx mori (B. mori) silkworm. Non-mulberry silkworms produce less silk than mulberry silkworms. Core silk fibrion (SF) and outer silk sericin (SS) make up the bulk of B. mori silk. In most cases, SF contributes between 60 and 80% of the overall cocoon weight, while SS contributes between 15 and 35%. The sericin layer also contains 1–5% nonsericin-like components, such as sugars, wax, and pigments. Textile production relies on silk. Recently, the field of biomedicine has become increasingly interested in silk as more is learned about its structure and properties [179].

Silk is an innovative biomaterial for tissue engineering that has been researched recently (Table 7). Silk has been examined because it is similar to fibrin/fibronectin in design, mechanical characteristics, and degradation rates but does not produce pathological hypertrophy [180]. Silk-based scaffolds exhibit therapeutic effects and sustain cardiac lineage differentiation in animal models [181]. In cardiac tissue studies, orientation is essential for sarcomere maintenance and development, notably titin protein overexpression [121,182]. Yeshi et al. (2021) proposed a Polypyrrole (PPy) scaffold coated with silk fibroin. Here, PPy was mixed with a silk fibroin (SF) solution to electrospin PPy-encapsulated SF nanofibers, which is an alternative to the more conventional aqueous coating. Electrospinning has been identified as a promising technique for producing nanofibrous mats imitating the structure of extracellular matrix (ECM) protein fibers [183].

Apart from nanofibers of silk derived from silkworms, other natural nanofibers are also present, for example, mussel-derived silk, spider-based nanofibers, and sea-silk-based nanofibers, snail-based nanofibers and have been used in several applications in tissue engineering [185].

## 4. Conclusions

Natural biodegradable nanopolymers (NAbioPOLY) have been shown to have enormous promise for use in cardiac tissue engineering, as this article elucidates. The benefits and drawbacks of various materials for cardiac tissue engineering are summarized in Table 8. These nanopolymers are at the forefront of cardiac tissue engineering because of their exceptional biocompatibility. Additional important features that have been studied and confirmed to be acceptable for cardiac applications are mechanical properties and rate of deterioration. The hydrogels made from naturally occurring matrices such as collagen, fibrinogen, alginate, chitosan, and silk are particularly well suited for cardiac repair thanks to their softness and flexibility. Hydrogels made from natural biomaterials have also been tested in a handful of clinical trials thus far. Moreover, another important application of NAbioPOLYs is the cardiac patches. According to the literature cited, it has been found that collagen I is mostly used for the development of cardiac patches along with fibrin and some synthetic nanopolymers. Collagen I is considered an appropriate material for the development of cardiac patches because of its high biocompatibility, low antigenicity, and adequate mechanical properties. Fibrin-based cardiac patches also depicted high biocompatibility. Another important application is the development of scaffolds, and NAbioPOLYs play a significant role in this. Collagen/chitosan-based scaffolds, gelatin-based scaffolds, alginate-based scaffolds, and silk-based scaffolds have been reported in the paper. All of these scaffolds depicted great biocompatibility as well as better mechanical stability. Eventually, in the preparation of bioinks, NabioPOLYs have also been used. For example, alginate-hyaluronic acid hydrogel bioinks have been reported that are extremely elastic, shear-thinning, biocompatible, and mechanically malleable.

To address the problems inherent in both natural and synthetic polymers, scientists have developed cutting-edge hybrid materials (a combination of both natural and synthetic polymers). Despite the progress made in cardiac tissue engineering, further research is still required for the development of more compatible materials that can be tested in-vivo as well as in-vitro. For this reason, efforts need to be made to improve cardiac tissue engineering by modifying the characteristics of biodegradable polymers, such as by adding components to a polymer to form a composite material.

## Figures and Tables

**Figure 1 nanomaterials-13-01374-f001:**
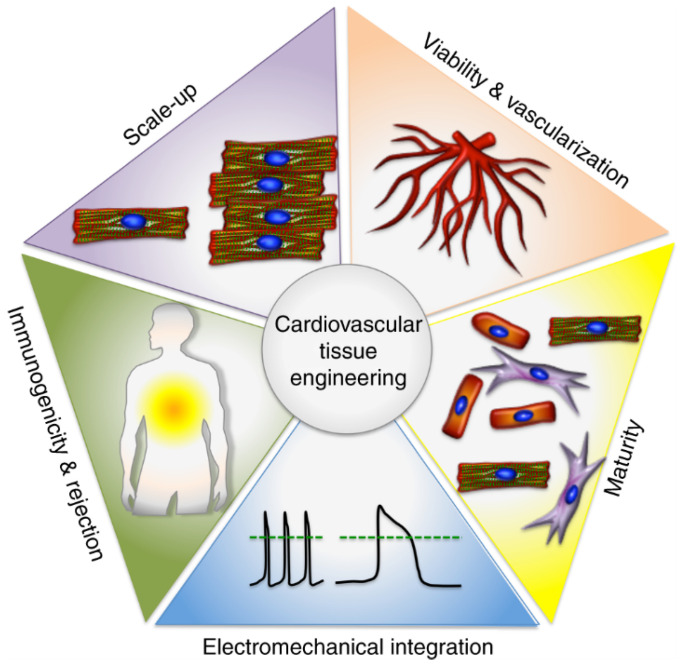
Cardiovascular tissue engineering bottlenecks.

**Figure 2 nanomaterials-13-01374-f002:**
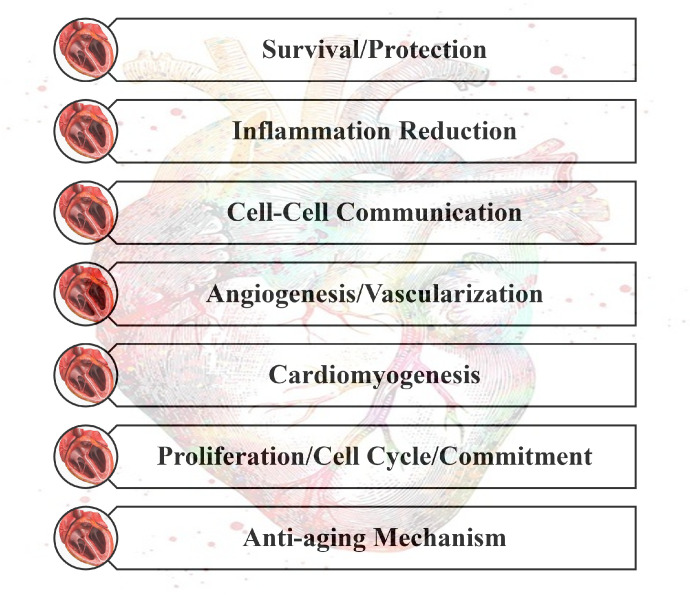
Mechanisms of cardiac repair and regeneration.

**Figure 3 nanomaterials-13-01374-f003:**
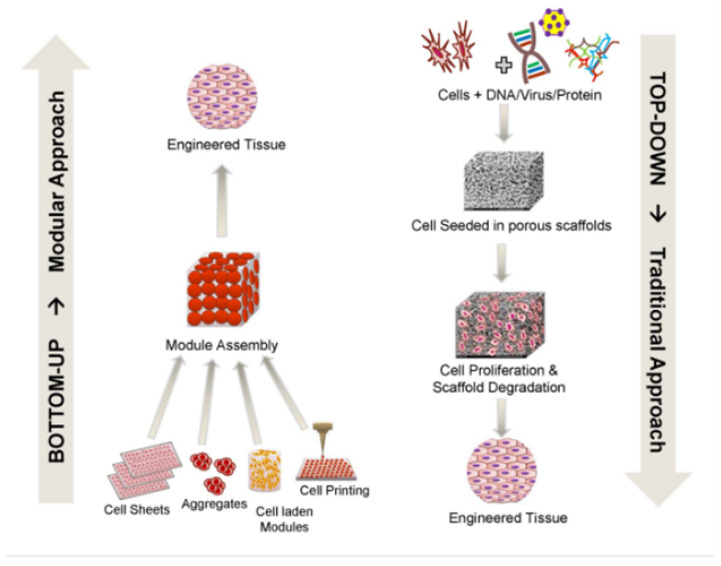
Comparison of top-down and bottom-up approaches for tissue engineering.

**Figure 4 nanomaterials-13-01374-f004:**
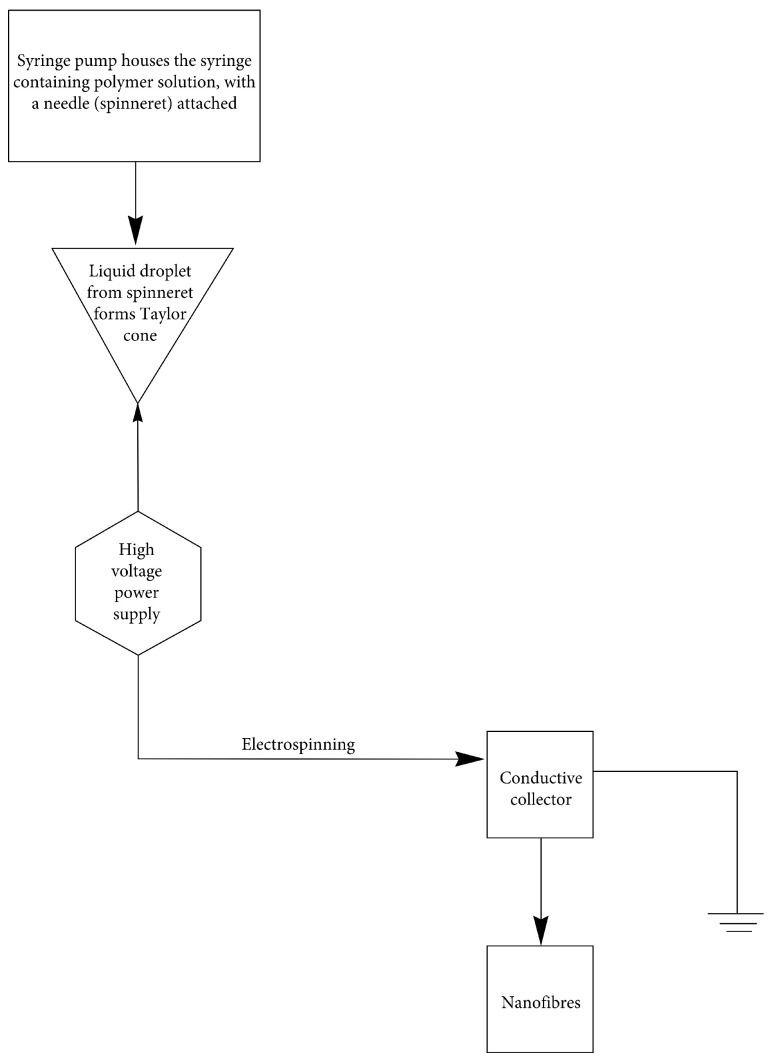
Schematic illustration of the mechanism of the traditional electrospinning method used to create nanofibers.

**Figure 5 nanomaterials-13-01374-f005:**
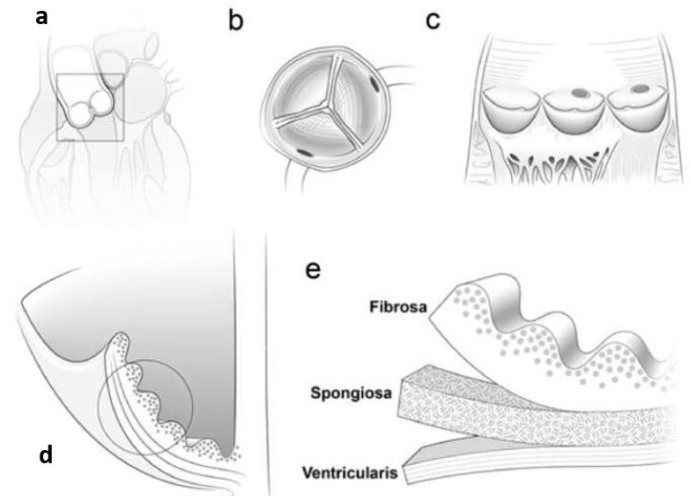
Aortic heart valve anatomy in schematic form. An aortic valve is shown in (**a**) a cross-sectional image of the heart. Illustration: (**b**) Aortic valve looking from above (seen from the aortic side). (**c**) View of the splayed-open valve from the side, revealing the semilunar shape of the cusps. (**d**) A sectional view through the cusp and aortic wall, revealing the cusp’s three-layer structure. (**e**) The collagen fibril orientations of the three cusp layers (fibrosa, spongiosa, and ventricularis).

**Figure 6 nanomaterials-13-01374-f006:**
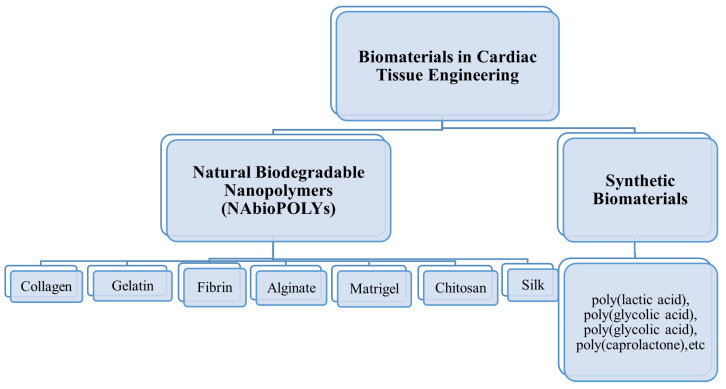
Types of biomaterials used in cardiac tissue engineering.

**Table 2 nanomaterials-13-01374-t002:** Gelatin-based scaffolds for heart tissue engineering.

Scaffold	Method of Biofabrication	Evaluated Properties	Source
Gelatin-based hydrogel	Chemical Cross-linking	Exhibit appropriate mechanical characteristics and excellent biocompatibility	[154]
Furfuryl-gelatin (f-gelatin) alone, f-gelatin with polycaprolactone (PCL) in a 1:1 ratio, and coaxial scaffolds with PCL (core) and f-gelatin (sheath)	Nozzle electrospinning, Coaxial electrospinning	The structural and mechanical stability of coaxial f-gelatin > PCL electrospun scaffolds of PCL (core) as well as f-gelatin (sheath) was superior to that of electrospun scaffolds created from f-gelatin alone and traditionally blended f-gelatin and PCL (1:1) scaffolds.	[155]

**Table 4 nanomaterials-13-01374-t004:** Alginate-based scaffolds for heart tissue engineering.

Scaffold	Method of Biofabrication	Evaluated Properties	Source
Magnetite nanoparticle-functionalized alginate scaffolds	Freeze-dry technique	The application of an alternating magnetic field on magnetic alginate scaffolds produces stimulating microenvironments useful in the engineering of functional tissues.	[165]
Alginate-chitosan cardiac ECM composite	Freezing and lyophilization	Porosity of over 96%, stability in PBS solution, very high swelling rate, increasing tensile strength, proliferation of human MSC inside pores, increased cTnT expression.	[166]
Alginate-hyaluronic acid hydrogel bioink	Chemical Cross-linking	Alg-HA gels were extremely elastic, shear-thinning, biocompatible, and mechanically malleable.	[167]

**Table 5 nanomaterials-13-01374-t005:** Chitosan-based scaffolds for heart tissue engineering.

Scaffold	Method of Biofabrication	Evaluated Properties	Source
Polyurethane/CS/carbon nanotubes composite	Electrospinning	Three advantages of aligned nanofibers are their biocompatibility, electrical conductivity, and potential for future applications.	[172]
Cardiac ECM-chitosan-gelatin composite	Freezing and lyophilization	Increased cell survival and proliferation, and facilitated differentiation were all achieved by using a high-porosity, biodegradable and biocompatible scaffold.	[173]
Chitosan-vitamin C based injectable hydrogel	Electrostatic adsorption	Appropriate gelation and injectability.	[174]

**Table 6 nanomaterials-13-01374-t006:** Matrigel-based scaffolds for heart tissue engineering.

Scaffold	Method of Biofabrication	Evaluated Properties	Source
First regenerated collagen/Matrigel-based ECTs in vitro and produced iron oxide nanoparticles (IRONs) with 2,3-dimercaptosuccinic acid (DMSA-IRONs).	Chemical Co-precipitation	DMSA-IRONs increased gap junctions and decreased heart cell mechanical connections (adherens junctions and desmosomes).	[177]

**Table 7 nanomaterials-13-01374-t007:** Silk-based scaffolds for heart tissue engineering.

Scaffold	Method of Biofabrication	Evaluated Properties	Source
Polypyrrole scaffold coated with silk fibroin	Electrospinning	Contraction of the ECM is supported by this mimic of myocardial fibrils, which has mechanical qualities similar to those of native myocardium and adequate electrical conductivity for cardiomyocytes.	[183]
Cellularized silk fibroin scaffold	Electrospinning	Producing high levels of cardiac differentiation markers in vitro with or without the use of fibroin meshes	[184]

**Table 8 nanomaterials-13-01374-t008:** Advantages and disadvantages of Natural biodegradable nanopolymers (NAbioPOLY) in cardiac tissue engineering.

NAbioPOLY	Advantages	Disadvantages	References
Collagen	It is more biocompatible than most natural polymers. Its cardiac cell attachment ligands make it bioactive. Modifiable biodegradability. Low antigenicity. Many physical, chemical, mechanical, and morphological features of collagen scaffolds can be tailored to achieve specific tasks. High-purity collagen can be obtained in big quantities from a variety of tissue sources at a minimal cost. It has many ligand sites to stimulate cardiac tissue regeneration. Many hard and soft tissues have ECMs made of collagen, especially fibrillar type I. Maintains myocyte alignment and matrix deformation resistance during the cardiac cycle, maintaining myocardial shape, thickness, and stiffness.	Gel-like materials’ low stiffness and lack of spatial bio-mimetic environment limit their in vivo uses. It is difficult to build collagen scaffolds with nonlinear elasticity like the heart muscle and beat synchronously with the receiver heart. The transplanted cardiac patch needs vascularization for mass transport, cell survival, electromechanical integration, and functional efficiency.	[124,126,137,152]
Gelatin	Abundantly found in the extracellular matrix of numerous organs, including the heart. Biocompatible.	Gelatin hydrogels have several limitations, including a lack of durability, mechanical stability, and high-water content.	[145,147,154]
Fibrin	Due to its better biocompatibility, predictable rate of degradation, and absence of toxicity, fibrin remained widely promoted and utilized in cardiovascular tissue engineering.	Including poor mechanical qualities, gel shrinkage, and the possibility of disease transmission.	[157,158,159]
Alginate	Alginates—natural polysaccharides— are biocompatible, biodegradable, non-toxic, non-immunogenic, and non-thrombogenic. Alginates are used in cell transplantation, medication and protein delivery, and wound healing.	Under physiological conditions, partial oxidation of alginate chains favors decomposition, but mammals lack the alginase enzyme, making alginate non-degradable.	[118,164]
	It can be intracoronarily or locally delivered into the infarcted myocardium, therefore it does not require open surgery.	Alginate hydrogels exhibit poor bioresorbability and cell adhesiveness, resulting in poor tissue contact and wound healing.	
Chitosan	Biocompatible, adhesive nature, and bactericidal as well as antifungal characteristics. Can be processed into films, membranes, hydrogels, fibers, scaffolds, and sponges.	Chitin dissolves poorly in conventional solvents due to its stiff crystalline structure. Chitin and chitosan come from crustaceans, insects, and fungi, causing batch-to-batch variation.	[169,174]
Matrigel	Cytocompatibility and inherent rigidity	Originating from murine sarcoma cell.	[120,123,176]
Silk	Good adherence with diverse tunability; bioresorbable; high elasticity.	Silk must be mixed with other materials for cardiac uses.	[66,178,181]

## Data Availability

The data that support the findings of this study are available from the corresponding author, R.A., upon reasonable request.

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
