# Peer review of "A Review on the Applications of Natural Biodegradable Nano Polymers in Cardiac Tissue Engineering"

_nanomaterials, 2023, doi:10.3390/nano13081374_

Round 1

Reviewer 1 Report

This review article can be accepted for publication after improvement. 

1. In the introduction section, some natural-based nanofibers should be introduced, for examples, mussel-derived silk, Latex-based Nanofibers, Silkworm-based silk,  spider-based nanofibers, and Sea-silk based  nanofibers,  Snail-based nanofibers. 

2. In the section “5.4. Electrospinning”, besides the electrospinning, the bubble electrospinning should be mentioned for mass-production of nanofibers. 

3. 3D printing technology for Cardiac Tissue Engineering should be also reviewed. Facta Universitatis Series: Mechanical Engineering (http://casopisi.junis.ni.ac.rs/index.php/FUMechEng ) published some articles on 3-D printing, which might be useful for the revision 

Author Response

Reviewer 1:

This review article can be accepted for publication after improvement. 

  1. In the introduction section, some natural-based nanofibers should be introduced, for examples, mussel-derived silk, Latex-based Nanofibers, Silkworm-based silk, spider-based nanofibers, and Sea-silk based nanofibers, Snail-based nanofibers. 

Response: We agree with the reviewer about the opportunity to introduce some natural-based nanofibers starting from the Section Introduction (see lines 70-81). Moreover, we used the reviewer’s suggestion to improve the paper by considering the relative literature in subsection 3.7 - Silk (lines 665-678).

  1. In the section “5.4. Electrospinning”, besides the electrospinning, the bubble electrospinning should be mentioned for mass-production of nanofibers. 

Response: Following the reviewer’s suggestion, Bubble Electrospinning has been added as Section 2.4.5.

  1. 3D printing technology for Cardiac Tissue Engineering should be also reviewed. Facta Universitatis Series: Mechanical Engineering (http://casopisi.junis.ni.ac.rs/index.php/FUMechEng ) published some articles on 3-D printing, which might be useful for the revision.

Response: Subsection 2.6 has been added according to the request, describing the 3D Bioprinting technology for Cardiac Tissue Engineering.

Reviewer 2 Report

In the framework of nanotechnology, nanomaterials have greatly facilitated the development of these cardiovascular stents because they offer easy tissue regeneration. Currently, functional nanofibers can be used for stem cell production and cell and tissue regeneration. I think the author's review is very valuable, but the author needs to make some revisions before accepting it.

1. The introduction is not enough. The author should focus on the research of functional nanofibers used in cell and tissue regeneration, and add more relevant literature.

What is the purpose of Section 2.2-4? The focus of the introduction is on functional nanofibers, so why add them?

3. The author seems to have no time to add relevant research, so I suggest adding time. This will make the article appear more organized.

Generally speaking, the author's writing seems to be incomplete, so it is suggested that the author add more literature to enrich the content of the article.

Author Response

Reviewer 2:

In the framework of nanotechnology, nanomaterials have greatly facilitated the development of these cardiovascular stents because they offer easy tissue regeneration. Currently, functional nanofibers can be used for stem cell production and cell and tissue regeneration. I think the author's review is very valuable, but the author needs to make some revisions before accepting it.

  1. The introduction is not enough. The author should focus on the research of functional nanofibers used in cell and tissue regeneration and add more relevant literature.

Response: We would like to thank the reviewer for the suggestion. Following that indication, the introduction has been extended with more details on biodegradable nanopolymers (NAbioPOLYs) along with the relevant literature review.

  1. What is the purpose of Section 2.2-4? The focus of the introduction is on functional nanofibers, so why add them?

Response: We would like to thank the reviewer for this observation that helped us to better organize the paper. Sections 2 to 4, together with section 5 and 6, were introduced by us to give a wide perspective on cardiac tissue engineering to the readers. Their relevance is quite high, as the main focus of the paper, that is on functional nanofibers. Nevertheless, the old organization of the paper did not reflect this aspect. Therefore, in the revised version of the paper,  the old 2 to 6 sections have been collected as a sub-section in the new Section 2 titled “ Perspective on cardiac tissue engineering”. 

  1. The author seems to have no time to add relevant research, so I suggest adding time. This will make the article appear more organized.

Response: We would like to thank the reviewer for the suggestion. In this new version of the manuscript, to consider as much relevant research as possible, we expanded the Introduction and introduced two more subsections, dealing with Bubble Electrospinning and 3D Bioprinting Technology for Cardiac Tissue Engineering (Section 2.6), respectively. Moreover, we better described all the subsections of (old Section 7.1) Section 3. In all the cases, we reported the relative bibliography in the references. 

Generally speaking, the author's writing seems to be incomplete, so it is suggested that the author add more literature to enrich the content of the article.

Response: Thank you for the comment. We hope that this new version of the manuscript could appear more complete thanks to the introduction of the new sections/subsections together with the cited bibliography (please see the response to point 3). 

Reviewer 3 Report

The review of Aziz et al. is devoated to the use of natural biodegradable nano polymers in cardiac tissue engineering. In addition to the main topic of the review Authors have discussed such topics as Cell Sources for Cardiac Tissue Engineering, Anatomy and Physiology of Human Heart, Regeneration of Cardiac CellsNanofabrication Approaches in Cardiac Tissue Engineering. In my opinion, this is the strength of the review, because it is the above aspects that determine the applicability of certain materials for tissue engineering. However, it is rather strange that Authors have paid more attetion on discussion of the topics described above than on the analysis of the use of natural biodegradable nano polymers in cardiac tissue engineering. I would like to suggest to expand the discussion of the use of natural biodegradable polymers. Adding an illustration would be beneficial too.

minor issue: in the caption to figure 5, the label "с" is missing

Author Response

Reviewer 3:

  1. The review of Aziz et al. is devoted to the use of natural biodegradable nano polymers in cardiac tissue engineering. In addition to the main topic of the review Authors have discussed such topics as Cell Sources for Cardiac Tissue Engineering, Anatomy and Physiology of Human Heart, Regeneration of Cardiac Cells,  Nanofabrication Approaches in Cardiac Tissue Engineering.
    In my opinion, this is the strength of the review, because it is the above aspects that determine the applicability of certain materials for tissue engineering. However, it is rather strange that Authors have paid more attention on discussion of the topics described above than on the analysis of the use of natural biodegradable nano polymers in cardiac tissue engineering. I would like to suggest to expand the discussion of the use of natural biodegradable polymers. Adding an illustration would be beneficial too.

Response: We thank the reviewer for this comment, and indeed, we expanded the discussion of the use of natural biodegradable polymers in Section 7 (now Section 3). Moreover, on the reviewer’s suggestion, an illustration as Figure 6 has been added, depicting the classification of biomaterials used in cardiac tissue engineering.

  1. minor issue: in the caption to figure 5, the label "с" is missing

Response: Thank you for pointing this out. Label "c" has been added to the caption of Figure 5.

Reviewer 4 Report

The review article on 26 pages contains 5 figures, 8 tables and 138 references.

The article provides an overview of natural biodegradable materials that are used in cardiovascular tissue engineering to develop heart patches, vessels and tissue. In addition, this article also provides an overview of cell sources for cardiac tissue engineering, anatomy and physiology of the human heart, cardiac cell regeneration, nanotechnology approaches used in cardiac tissue engineering as well as in scaffolds.

CONTENT:

1. The content of the work to a certain extent corresponds to the subject and profile of the journal.

2. The originality and novelty of the material proposed by the authors of the article are not at a high level. The analyzed literature sources include publications from 1992 to 2022, most of which are from 2008-2022.

3. The theoretical and practical significance of the work is due to the possibility of using the results obtained in the article in the conduct of research and development related to the production of effective biodegradable materials.

4. The quality of the analysis of the problem and the validity of its relevance in the introductory part of the manuscript are at an acceptable level.

5. The problem posed in the article has not been completely solved.

6. The information provided in the article is quite correct and its discussion was carried out in accordance with existing scientific concepts. Possible discussion points are within the framework adopted for scientific articles.

7. The sequence of presentation complies with the standard requirements of scientific journals.

8. The literacy of the presentation is good, the style of presentation corresponds to the scientific nature of the material, and the terminology used is accepted.

9. The design of the article and its volume are quite normal, the figures and tables correspond to the topic being presented and are appropriate for illustrating the text, most of the literary sources are recent publications

Notes:

1) The introduction of the abbreviation "NAbioPOLY" (Natural Biodegradable Nanopolymers) in the article is uncommon and redundant.

2) In section “7. Biomaterials Used In Cardiac Tissue Engineering” there is only paragraph “7.1. Natural Biodegradable Polymers (NAbioPOLY)”, and there is no paragraph 7.2. Perhaps the whole section should be called “7. Natural Biodegradable Polymers" without further crushing.

3) In paragraph “7.1. Natural Biodegradable Polymers (NAbioPOLY)" reviewed material 7.1.6. Matrigel, which should be added to the summary Table 8, indicating the advantages and disadvantages

4) In Table 8, it is necessary to provide references to literary sources for each material.

5) Sections with “2. Cell Sources for Cardiac Tissue Engineering2" to "6. Scaffolds for Cardiovascular Tissue Engineering" make up half of the volume of the article, but at the same time they are of a nature that is not directly related to the topic of the review, and are more suitable for section "1. Introduction". It is necessary to reduce them as much as possible and focus on increasing paragraph “7.1. Natural Biodegradable Polymers (NAbioPOLY)”, which reflects the essence of the review. Perhaps section "5. Nanofabrication Approaches in Cardiac Tissue Engineering”, describing various methods for obtaining materials, it is necessary to devote more to the topic of the article, to analyze the application of these methods to obtain Natural Biodegradable Nanopolymers.

6) Section “7.1. Natural Biodegradable Polymers (NAbioPOLY)" needs to be expanded as the information provided is rather sparse and has been presented to a similar extent in more general reviews of materials in Cardiac Tissue Engineering. For example, Mohammadi Nasr S, Rabiee N, Hajebi S, Ahmadi S, Fatahi Y, Hosseini M, Bagherzadeh M, Ghadiri AM, Rabiee M, Jajarmi V, Webster TJ. Biodegradable Nanopolymers in Cardiac Tissue Engineering: From Concept Towards Nanomedicine. Int J Nanomedicine. 2020 Jun 18;15:4205-4224. doi: 10.2147/IJN.S245936. PMID: 32606673; PMCID: PMC7314574.

CONCLUSION: The article can be accepted for publication after editing.

Author Response

Reviewer 4:

The review article on 26 pages contains 5 figures, 8 tables and 138 references. The article provides an overview of natural biodegradable materials that are used in cardiovascular tissue engineering to develop heart patches, vessels and tissue. In addition, this article also provides an overview of cell sources for cardiac tissue engineering, anatomy and physiology of the human heart, cardiac cell regeneration, nanotechnology approaches used in cardiac tissue engineering as well as in scaffolds.

CONTENT:

  1. The content of the work to a certain extent corresponds to the subject and profile of the journal.
  2. The originality and novelty of the material proposed by the authors of the article are not at a high

level. The analyzed literature sources include publications from 1992 to 2022, most of which are from 2008-2022.

  1. The theoretical and practical significance of the work is due to the possibility of using the results

obtained in the article in the conduct of research and development related to the production of

effective biodegradable materials.

  1. The quality of the analysis of the problem and the validity of its relevance in the introductory part of the manuscript are at an acceptable level.
  2. The problem posed in the article has not been completely solved.
  3. The information provided in the article is quite correct and its discussion was carried out in

accordance with existing scientific concepts. Possible discussion points are within the framework

adopted for scientific articles.

  1. The sequence of presentation complies with the standard requirements of scientific journals.
  2. The literacy of the presentation is good, the style of presentation corresponds to the scientific nature of the material, and the terminology used is accepted.
  3. The design of the article and its volume are quite normal, the figures and tables correspond to the

topic being presented and are appropriate for illustrating the text, most of the literary sources are

recent publications

Notes:

1) The introduction of the abbreviation "NAbioPOLY" (Natural Biodegradable Nanopolymers) in the article is uncommon and redundant.

Response: Following your suggestion, more explanations about the NAbioPOLY have been added to the Section Introduction as well as old Section 7.1 (new Section 3). 

2) In section “7. Biomaterials Used In Cardiac Tissue Engineering” there is only paragraph “7.1. Natural Biodegradable Polymers (NAbioPOLY)”, and there is no paragraph 7.2. Perhaps the whole section should be called “7. Natural Biodegradable Polymers" without further crushing.

Response: We would like to thank the reviewer for this observation that helped us to understand the missing emphasis that we gave on the main part of the paper, represented by this specific paragraph. We fully agree on the redundancy of the section numbering,  therefore, we adopted, for this section, the title “Natural Biodegradable Nanopolymers Used in Cardiac Tissue Engineering” and we made corrections on the remaining part of the paragraph in order to follow this idea. Moreover, we would also like to point out that the old Section 7 is now Section 3.

3) In paragraph “7.1. Natural Biodegradable Polymers (NAbioPOLY)" reviewed material 7.1.6. Matrigel, which should be added to the summary Table 8, indicating the advantages and disadvantages.

Response: Thank you for pointing this out. The missing information on the advantages and disadvantages of matrigel have now been added to Table 8.

4) In Table 8, it is necessary to provide references to literary sources for each material.

Response: We would like to thank the reviewer for this observation that helped us to rightly refer to the analyzed literature. According to her/his suggestion, a new column has been introduced reporting the reference number.  

5a) Sections with “2. Cell Sources for Cardiac Tissue Engineering2" to "6. Scaffolds for Cardiovascular Tissue Engineering" make up half of the volume of the article, but at the same time they are of a nature that is not directly related to the topic of the review, and are more suitable for section "1. Introduction". It is necessary to reduce them as much as possible and focus on increasing paragraph “7.1. Natural Biodegradable Polymers (NAbioPOLY)”, which reflects the essence of the review.

Response: We would like to thank the reviewer for this observation helping to better organize the paper. As he/she puts in evidence, sections from 2 to 6 seem to be information suitable for the “introduction” section. Nevertheless, they have been introduced by us to give a wide perspective on cardiac tissue engineering to the readers. Their relevance is quite high as the main focus of the paper, as also the comments given us from the other reviewers confirmed (they all asked to go into the details of some parts of these sections). Therefore, in the revised version of the paper a Section “2. Perspective on cardiac tissue engineering” composed by the old 2 to 6 Section has been introduced, and the details here reported have been reduced where it has been possible. Moreover, according to this suggestion, we also increased paragraph 7.1 (now Section 3).

5b)  Perhaps section "5 Nanofabrication Approaches in Cardiac Tissue Engineering”, describing various methods for obtaining materials, it is necessary to devote more to the topic of the article, to analyze the application of these methods to obtain Natural Biodegradable Nanopolymers.

Response: We would like to thank the reviewer for this observation. As described in the previous answer, section "5 Nanofabrication Approaches in Cardiac Tissue Engineering” has been introduced in order to give a general overview on the nanofabrication approaches used in Cardiac tissue engineering, and now it is part of the Section  “2. Perspective on cardiac tissue engineering”, whereas more accurate and focused  information on the application of these methods to obtain Natural Biodegradable Nanopolymers are provided in a dedicated Section on Nabiopolys (see Section 3). In order to clarify this aspect, the text of the old Section 5 has been improved.

Introduction of 2.4 Text:  A general overview on the nanofabrication approaches used in Cardiac tissue engineering is here given, whereas more accurate and focused  information on the application of these methods to obtain Natural Biodegradable Nanopolymers are provided in a dedicated Section (see Section 3).

6) Section “7.1. Natural Biodegradable Polymers (NAbioPOLY)" needs to be expanded as the information provided is rather sparse and has been presented to a similar extent in more general reviews of materials in Cardiac Tissue Engineering. For example, Mohammadi Nasr S, Rabiee N, Hajebi S, Ahmadi S, Fatahi Y, Hosseini M, Bagherzadeh M, Ghadiri AM, Rabiee M, Jajarmi V, Webster TJ. Biodegradable Nanopolymers in Cardiac Tissue Engineering: From Concept Towards Nanomedicine. Int J Nanomedicine. 2020 Jun 18;15:4205-4224. doi: 10.2147/IJN.S245936. PMID: 32606673; PMCID: PMC7314574.

Response: We thank the reviewer for the comment. Following his/her suggestion, we expanded the old Section 7.1 (now Section 3) adding more details on NAbiopolys. Also, we added a discussion of Nabiopolys in the introduction as well.  Moreover, we would like to point out that the suggested paper has been already considered in the original submission as paper [old ref?] and it inspired us. It was published in 2020. With respect to this, literature up to 2022 has been analyzed.

CONCLUSION: The article can be accepted for publication after editing.

Round 2

Reviewer 2 Report

I agree to accept it.

Author Response

Reviewer 1:

I agree to accept it.

Response: Thank you so much for your approval of our manuscript. Your suggestions have been really useful to improve it.

Reviewer 3 Report

The Authors have done a good job of improving the text of the review. The addition to the introduction and the division of the review into two semantic parts, in my opinion, made the text more clear for the reader. I believe that the review can be accepted for publication after a few small remarks are taken into account: 1) line 328: missing word at the beginning of a sentence 2) I recommend to reedit the paragraph 3.3. Now the two parts of the paragaraf, namely the original text and the added one, are poorly combined with each other. So in the original text it appears twice that fibrin is widely used for tissue engineering, the added text begins with the third mention of this fact. In addition, the original text first describes the pros and then the cons of fibrin, after which the description of the pros is added again, it looks a little strange.

Author Response

The Authors have done a good job of improving the text of the review. The addition to the introduction and the division of the review into two semantic parts, in my opinion, made the text more clear for the reader. 

Response: Thank you so much for your appreciation of the revised version of our manuscript. Your suggestions have been really useful to improve it.

I believe that the review can be accepted for publication after a few small remarks are taken into account: 

1) line 328: missing word at the beginning of a sentence.

Response: Thank you for pointing this out. Line 328 has been amended and it started as “It is used to …”

2) I recommend to reedit the paragraph 3.3. Now the two parts of the paragaraf, namely the original text and the added one, are poorly combined with each other. So in the original text it appears twice that fibrin is widely used for tissue engineering, the added text begins with the third mention of this fact. In addition, the original text first describes the pros and then the cons of fibrin, after which the description of the pros is added again, it looks a little strange.

Response: Thank you so much for the suggestion. We agree with your observation and following your suggestion, we reduced the redundancy of the phrase i.e., fibrin is widely used for cardiac tissue engineering. Moreover, we re-edited Section 3.3 in such a way that we combined all the advantages of fibrin and respective applications together and we explained its disadvantages in the separate paragraph. 

All the modifications are highlighted in yellow.